# Two Birds, One Stone: Achieving Differential Privacy and Certified Robustness for Pre-Trained Classifiers via Input Perturbation

## Abstract

Recent studies have shown that pre-trained classifiers are increasingly powerful to improve the performance on different tasks, e.g, neural language processing, image classification. However, adversarial examples from attackers can trick pre-trained classifier to misclassify. To solve this challenge, a reconstruction network is built before the public pre-trained classifiers to offer certified robustness and defend against adversarial examples through input perturbation. On the other hand, the reconstruction network requires training on the dataset, which incurs privacy leakage of training data through inference attacks. To prevent this leakage, differential privacy (DP) is applied to offer a provable privacy guarantee on training data through gradient perturbation. Most existing works employ certified robustness and DP independently and fail to exploit the fact that input perturbation designed to achieve certified robustness can achieve (partial) DP. In this paper, we propose perturbation transformation to show how the input perturbation designed for certified robustness can be transformed into gradient perturbation during training. We propose Multivariate Gaussian mechanism to analyze the privacy guarantee of this transformed gradient perturbation and precisely quantify the level of DP achieved by input perturbation. To satisfy the overall DP requirement, we add additional gradient perturbation during training and propose Mixed Multivariate Gaussian Analysis to analyze the privacy guarantee provided by the transformed gradient perturbation and additional gradient perturbation. Moreover, we prove that Mixed Multivariate Gaussian Analysis can work with moments accountant to provide a tight DP estimation. Extensive experiments on benchmark datasets show that our framework significantly outperforms state-of-the-art methods and achieves better accuracy and robustness under the same privacy guarantee.

## 1 Introduction

Deep learning with pre-trained classifiers are increasingly powerful for solving difficult machine-learning tasks in the real-world, including image classification (Krizhevsky et al., 2012; Simon et al., 2016) and natural language processing (Vaswani et al., 2017; Devlin et al., 2018). However, deep learning models with pre-trained classifiers are subject to adversarial examples attacks (Szegedy et al., 2013; Bruna et al., 2013; Goodfellow et al., 2014) which apply small perturbations on inputs to cause the model to misclassify. To solve this challenge, certified robustness (Li et al., 2018) for pre-trained classifiers is proposed by (Salman et al., 2020) as a provable defense approach, where a denoiser, e.g., autoencoder (AE) (Hinton & Zemel, 1994), is trained on the data perturbed with Gaussian noise and aims to reconstruct denoised data. Once denoiser finishes training, it takes the Gaussian perturbed data for denoising reconstruction and feeds the denoised data to the pre-trained classifier. Randomized smoothing is then applied and provides certified robustness without retraining the large pre-trained model.

On the other hand, the training of denoiser requires training data that can include sensitive information, e.g., clinical records, financial records, user profiles, etc. Several works have shown that attackers can infer private information from training data through trained models (Fredrikson et al., 2015; Wang et al., 2015; Shokri et al., 2017). A popular and powerful technique to address this issue in deep learning is differential privacy (DP) (Dwork et al., 2006; Dwork, 2011; Dwork et al., 2014),

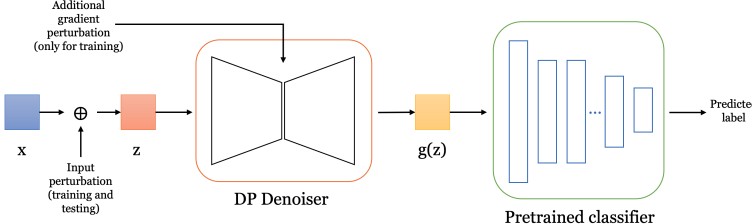

Figure 1: Framework of TransDenoiser: given a clean image **x**, input perturbation is added to generate perturbed image **z**. **z** is then reconstructed by denoiser $g$ to generate $g(\mathbf{z})$, which is fed into the pre-trained classifier $h$ for classification. The input perturbation on **x** is utilized to achieve certified robustness during testing. The denoiser is trained under DP by leveraging the input perturbation added on **x** and additional gradient perturbation during training.

which has been adopted in a large volume of works to provide a rigorous protection from leaking private information contained in training data. Gaussian Mechanism (**GM**) (Dwork et al., 2014) is a basic technique to achieve DP by injecting isotropic Gaussian perturbations to a computation output. Depending on where to inject the perturbations, existing works on machine learning with DP can be mainly categorized into: input perturbation (Fukuchi et al., 2017; Kang et al., 2020a;b), output perturbation (Zhang et al., 2017), gradient perturbation (Song et al., 2013; Bassily et al., 2014; Shokri & Shmatikov, 2015; Abadi et al., 2016; 2017; Wang et al., 2017; Lee & Kifer, 2018; Yu et al., 2019), objective perturbation (Kifer et al., 2012; Zhang et al., 2012; Phan et al., 2016; 2017; Iyengar et al., 2019), and noisy labelling (Papernot et al., 2016; 2018).

Therefore, a straightforward way to prevent these two critical risks, i.e., adversarial examples and privacy leakage, is independently applying certified robustness and DP to models with pre-trained classifiers. A few works (Phan et al., 2019; 2020) follow this idea and simultaneously achieve both DP and certified robustness. However, we observe that the randomized smoothing for certified robustness requires that the input data is perturbed with Gaussian noise during training. This Gaussian perturbation brings randomization to the training process and could have been utilized to provide a certain level of DP guarantee. Independently applying certified robustness and DP fails to exploit this connection and incurs unnecessary additional randomization during training, which leads to the degradation of model utility.

Directly analyzing the DP guarantee through input perturbation methods (Fukuchi et al., 2017; Kang et al., 2020a;b) is nontrivial in deep learning, because these methods pose strict constraints on loss function, which are not satisfied by deep learning models. An alternative way is to transform the input perturbation into gradient perturbation and then analyze the DP guarantee it provides. However, existing work (Kang et al., 2020a) gives strong assumption and simply regards the transformed gradient perturbation as isotropic Gaussian perturbation, and fails to recognize that in most deep learning models, this transformed gradient perturbation follows multivariate Gaussian distribution.

In this paper, we propose a novel framework TransDenoiser (Figure 1) to simultaneously achieve certified robustness and DP for models with pre-trained classifiers. TransDenoiser has a similar architecture as (Salman et al., 2020) by adding a denoiser before pre-trained classifier. Compared with (Salman et al., 2020), TransDenoiser can provide similar level of certified robustness without retraining the pre-trained model, as well as guarantee DP for the training data. Compared with existing works that achieve both certified robustness and DP including SecureSGD (Phan et al., 2019) and StoBatch (Phan et al., 2020), TransDenoiser 1) provides a tigher guarantee of DP by utilizing all the randomization during training including input and gradient perturbations, and 2) achieves more effective certified robustness by leveraging randomized smoothing on the input instead of noisy layers in the model.

**Contributions.** Our key contributions are:

1. We propose a novel framework TransDenoiser that trains a denoiser through both input and gradient perturbation for achieving DP and certified robustness simultaneously on deep learning models with pre-trained classifiers. The input perturbation for achieving certified robustness is utilized to achieve partial DP and additional gradient perturbation is used as necessary for the overall DP, ensuring an enhanced privacy and utility performance.

2. We present an analytical tool that leverages Taylor expansion to transform input perturbation into gradient perturbation so that it can be quantified and composed with the explicit gradient perturbation for the DP guarantee. We propose a Multivariate Gaussian Mechanism (**MGM**) to analyze DP of the multivariate Gaussian perturbation and prove that **MGM** is a generalization of Heterogeneous Gaussian Mechanism (Phan et al., 2019).

3. Observing that the transformed gradient perturbation itself cannot satisfy the DP guarantee requirement in some scenario, we add additional perturbation following isotropic Gaussian distribution to the gradient, and propose Mixed Multivariate Gaussian Analysis (**MMGA**) to analyze the DP guarantee provided by transformed gradient perturbation and additional gradient perturbation. We also prove that **MMGA** can work with moments accountant (Abadi et al., 2016) to provide a tight bound on the privacy cost.

4. We conduct extensive experiments on several benchmark datasets which demonstrate that TransDenoiser can 1) provide a significantly tighter bound on privacy cost with same utility performance, and 2) achieve similar level of certified robustness as other state-of-the-art works.

## 2 TRANSDENOISER

In this section, we will present our proposed framework TransDenoiser, which can be applied before any pre-trained classifier to guarantee certified robustness and DP without retraining the pre-trained model. A denoiser is trained to guarantee certified robustness of the final model via randomized smoothing or input perturbation on the input, where the training process introduces the input perturbation for better accuracy of the randomized smoothed model. We transform the input perturbation into equivalent gradient perturbation so that it can be quantified and composed with the explicit gradient perturbation for the DP guarantee.

### 2.1 DENOISER AND CERTIFIED ROBUSTNESS

As can be seen in Figure 1, TransDenoiser is a denoising AE trained on input data with Gaussian perturbation. Similar to (Salman et al., 2020), this input Gaussian perturbation is used as randomized smoothing for certified robustness. Naively applying randomized smoothing on a pre-trained classifier without the denoiser gives very loose certification bounds because the pre-trained classifier is not trained to be robust to Gaussian perturbations of their input. The denoiser serves to "remove" this Gaussian perturbation and effectively reconstruct the input data like a pre-processing step before feeding input data into the pre-trained classifier while maintaining the benefit of certified robustness. The detailed proof of randomized smoothing can be found in (Cohen et al., 2019), and we provide a brief proof in Appendix B.

Different from (Salman et al., 2020), given input data $\mathbf{x}_{(i)}$ and perturbed data $\mathbf{z}_{(i)} = \mathbf{x}_{(i)} + \mathbf{b}_{(i)}$ with $\mathbf{b}_{(i)} \sim \mathcal{N}(0, \sigma^2 I)$, the objective function we use to optimize the denoiser contains the standard reconstruction MSE:

$$l(\mathbf{z}_{(i)}, \theta) = \|g(\mathbf{z}_{(i)}) - \mathbf{x}_{(i)}\|_2^2, \tag{1}$$

where $l$ is the loss function, $g$ denotes the denoiser, and $\theta$ is the parameter of the denoiser. The stability objective of (Salman et al., 2020) is not included in our objective function, because it requires both $\mathbf{x}_{(i)}$ and $g(\mathbf{z}_{(i)})$ to pass the pre-trained classifier $h$, which both incurs additional privacy cost and additional computation overhead when we calculate the transformation matrix. In this work, we assume $l(\mathbf{z}_{(i)}, \theta)$ is $C$-Lipschitz continuous, which is a mild and common assumption in existing works (Bassily et al., 2019; Feldman et al., 2020).

### 2.2 PERTURBATION TRANSFORMATION AND MULTIVARIATE GAUSSIAN MECHANISM

In this section, we will introduce perturbation transformation and Multivariate Gaussian Mechanism (**MGM**) to analyze the DP guarantees of the input perturbation.

**Perturbation transformation.** As we introduced in 2.1, input Gaussian perturbation is utilized to achieve certified robustness for TransDenoiser. Although theoretically this perturbation is only required at the testing phase, almost all existing approaches in practice demand the randomization

during training to improve the performance. Our strategy is to transform input perturbation into gradient perturbation during training and analyze the DP guarantee that input perturbation can offer. The crucial step of this transformation is the Taylor expansion of MSE loss $l(\mathbf{z}_{(i)}, \theta)$ at the data point $\mathbf{x}_{(i)}$, which is formulated as follows,

$$l(\mathbf{z}_{(i)}, \theta) = l(\mathbf{x}_{(i)}, \theta) + (\mathbf{z}_{(i)} - \mathbf{x}_{(i)})^{\mathsf{T}} \nabla_{\mathbf{x}_{(i)}} l(\mathbf{x}_{(i)}, \theta) + o(\mathbf{z}_{(i)} - \mathbf{x}_{(i)}) \tag{2}$$

Since the only mild constraint on $l(\mathbf{z}_{(i)}, \theta)$ is $C$-Lipschitz continuous, it is possible for the higher order terms $o(\mathbf{z}_{(i)} - \mathbf{x}_{(i)})$ to be negative. We denote the examples with non-negative higher order terms as "non-negative cases", and the others as "negative cases". In the rest of this section, we use the superscript "non" and "neg" to denote samples of "non-negative cases" and "negative cases", respectively. For "non-negative cases", we have the following lemma:

**Lemma 1.** *Given perturbed example $\mathbf{z}_{(i)}^{non} = \mathbf{x}_{(i)}^{non} + \mathbf{b}_{(i)}$ with $\mathbf{b}_{(i)}^{(k)} \sim \mathcal{N}(0, \sigma^2)$, and MSE loss $l(\mathbf{z}_{(i)}^{non}, \theta)$ is C-Lipschitz continuous. The gradient $\nabla_\theta l(\mathbf{z}_{(i)}^{non}, \theta)$ can be reformulated as the gradient with respect to the original sample with a gradient perturbation:*

$$\nabla_\theta l(\mathbf{z}_{(i)}^{non}, \theta) \geq \nabla_\theta l(\mathbf{x}_{(i)}^{non}, \theta) + \mathbf{p}_{(i)}, \tag{3}$$

*where $\mathbf{p}_{(i)}$ is the transformed perturbation with $\mathbf{p}_{(i)} \sim \mathcal{N}(0, \boldsymbol{\Sigma}_{(i)})$, $\boldsymbol{\Sigma}_{(i)} = \mathbf{M}_{(i)} \sigma^2$, $\mathbf{M}_{(i)} = \mathbf{A}_{(i)} \mathbf{A}_{(i)}^{\mathsf{T}}$ and $\mathbf{A}_{(i)} = \mathbf{J}_\theta \nabla_{\mathbf{x}_{(i)}^{non}} l(\mathbf{x}_{(i)}^{non}, \theta)$.*

The detailed proof of Lemma 1 can be found in Appendix E. With Lemma 1, we find that the right-hand side is the lower bound of left-hand side, which means DP guarantee provided by transformed gradient perturbation $\mathbf{p}_{(i)}$ is the lower bound of that provided by input perturbation $\mathbf{b}_{(i)}$.

**Multivariate Gaussian Mechanism.** Since the transformed gradient perturbation $\mathbf{p}_{(i)}$ in equation (3) follows a multivariate Gaussian distribution with correlated elements, which is in contrast to the independent and isotropic Gaussian noise used in standard Gaussian mechanism for DP, we introduce Multivariate Gaussian Mechanism (**MGM**) to analyze DP of this multivariate Gaussian perturbation.

**Theorem 1.** *Multivariate Gaussian Mechanism. Let $\mathcal{G} : \mathbb{R}^v \to \mathbb{R}^w$ be an arbitrary $w$-dimensional function, and $\Delta_{\mathcal{G}} = \max_{\mathcal{D}, \mathcal{D}'} \|\mathcal{G}(\mathcal{D}) - \mathcal{G}(\mathcal{D}')\|_2$. A Multivariate Gaussian Mechanism $\mathcal{M}$ with the covariance $\boldsymbol{\Sigma} \in \mathbb{R}^{w \times w}$ adds noise to each of the $w$ elements of the output. The mechanism $\mathcal{M}$ is $(\epsilon, \delta)$-DP, with*

$$\epsilon \in (0, 1], \ S_{min}(\mathbf{M})^{\frac{1}{2}} \sigma \geq \sqrt{2 \ln(1.25/\delta)} \Delta_{\mathcal{G}} / \epsilon.$$

*where $S_{min}(\mathbf{M})$ is the minimum singular value of $\mathbf{M}$ and $\boldsymbol{\Sigma} \triangleq \mathbf{M} \sigma^2$.*

The proof of this theorem is in Appendix D. With Theorem 1, multivariate Gaussian perturbation can be leveraged to preserve $(\epsilon, \delta)$-DP, and we have the following Corollary:

**Corollary 1.** *Given a multivariate Gaussian perturbation $\mathbf{p} \sim \mathcal{N}(0, \boldsymbol{\Sigma})$, $\boldsymbol{\Sigma} = \mathbf{M} \sigma^2$, the DP guarantee of the **MGM** is equivalent to that of a **GM** with its perturbation following a Gaussian distribution $\mathcal{N}(0, S_{min}(\mathbf{M}) \sigma^2)$, where $S_{min}(\mathbf{M})$ is the minimum singular value of $\mathbf{M}$.*

Because the DP guarantee of **MGM** is equivalent to the **GM** by applying a transformed perturbation, the traditional DP analysis technique, e.g., moments accountant Abadi et al. (2016), can be leveraged to analyze the privacy cost in the training process of denoiser. We also note a special case of **MGM**, in which the covariance matrix of the multivariate Gaussian perturbation only contains the diagonal values, and the perturbation on each elements is independent from each other but they can have different scales. This mechanism is called Heterogeneous Gaussian Mechanism (**HGM**) Phan et al. (2019). We re-define **HGM** in Appendix F and prove that **MGM** is a generalization of **HGM**.

## 2.3 TRANSDENOISER TRAINING ALGORITHM

As introduced in Section 2.2, the DP guarantee provided by transformed gradient perturbation depends on the transformation matrix and the scale of input noise. In some scenarios, this transformed gradient perturbation itself does not fully satisfy the DP requirement, because the scale of input noise

to achieve randomized smoothing of certified robustness is relatively small. To address this, we add additional gradient perturbation directly to the gradient in each iteration of the training process.

Algorithm 1 shows the details of our proposed TransDenoiser training algorithm to achieve both certified robustness and DP. Each record is perturbed with input perturbation to achieve certified robustness (line 7). We utilize both input perturbation and gradient perturbation to achieve DP for "non-negative cases" (Line 12 - 18), and for the "negative cases", we directly employ gradient perturbation (Line 19 - 23). For non-negative cases, we transform input perturbation at each iteration into gradient perturbation. We then add additional gradient perturbation with scale $\bar{\sigma}$ to $\nabla_{\theta_t} l(\mathbf{z}_t, \theta_t)$ at each iteration given hyper-parameters $\xi_{low}$ and $\xi_{up}$. In Algorithm 1, we use the commonly used approach, mini-batch SGD, to train the denoiser, which is slightly different from our previous setting (Lemma 1) where only a single sample is fed into the model per iteration. We will show that our DP analysis works in both of these two settings.

---

**Algorithm 1:** TransDenoiser Training Algorithm

---

**Input:** pre-trained classifier $h$, total training epoch $T$, perturbation scale thresholds $\xi_{low}$ and $\xi_{up}$, input perturbation scale $\sigma$, learning rate $\eta$, training dataset $D_{train}$

1  $t = 0$;
2  load parameters of the pre-trained classifier $h$;
3  initialize parameters of the denoiser $g$;
4  **while** $t < T$ **do**
5      get mini-batch data $\mathbf{x}_t$ from $D_{train}$;
6      **for** *each data* $\mathbf{x}_{(i)}$ *in* $\mathbf{x}_t$ **do**
7          $\mathbf{z}_{(i)} := \mathbf{x}_{(i)} + \mathcal{N}(0, \sigma^2)$;
8          $o(\mathbf{z}_{(i)} - \mathbf{x}_{(i)}) := l(\mathbf{z}_{(i)}, \theta) - (l(\mathbf{x}_{(i)}, \theta) + (\mathbf{z}_{(i)} - \mathbf{x}_{(i)})^{\mathsf{T}} \nabla_{\mathbf{x}_{(i)}} l(\mathbf{x}_{(i)}, \theta))$;
9      **end**
10     "Non-negative cases" $\mathbf{z}_t^{non} := \{\mathbf{z}_{(i)}\}$ with $o(\mathbf{z}_{(i)} - \mathbf{x}_{(i)}) \geq 0$;
11     "Negative cases" $\mathbf{z}_t^{neg} := \{\mathbf{z}_{(i)}\}$ with $o(\mathbf{z}_{(i)} - \mathbf{x}_{(i)}) < 0$;
12     **if** *"Non-negative cases"* **then**
13         compute the gradient $\nabla_{\theta_t} l(\theta_t, \mathbf{z}_t^{non})$;
14         calculate the input perturbation transformation matrix $\mathbf{M}_t$;
15         calculate $S_{min}(\mathbf{M}_t)$;
16         $\bar{\sigma} := \begin{cases} \xi_{up}, & \text{if } \sqrt{TS_{min}(\mathbf{M}_t)}\sigma < \xi_{low}, \\ \sqrt{\xi_{up}^2 - TS_{min}(\mathbf{M}_t)\sigma^2}, & \text{else if } \sqrt{TS_{min}(\mathbf{M}_t)}\sigma < \xi_{up}, ; \\ 0, & \text{else}, \end{cases}$
17         add additional perturbation to the gradient $\nabla_\theta l(\theta_t, \mathbf{z}_t^{non}) := \nabla_\theta l(\theta_t, \mathbf{z}_t^{non}) + \mathcal{N}(0, \bar{\sigma}^2)$;
18     **end**
19     **else**
20         compute the gradient $\nabla_{\theta_t} l(\theta_t, \mathbf{z}_t^{neg})$;
21         $\bar{\sigma} := \xi_{up}$;
22         add perturbation to the gradient $\nabla_\theta l(\theta_t, \mathbf{z}_t^{neg}) := \nabla_\theta l(\theta_t, \mathbf{z}_t^{neg}) + \mathcal{N}(0, \bar{\sigma}^2)$;
23     **end**
24     $\nabla_{\theta_t} l(\theta_t, \mathbf{z}_t) := \nabla_\theta l(\theta_t, \mathbf{z}_t^{non}) + \nabla_\theta l(\theta_t, \mathbf{z}_t^{neg})$ ;
25     update parameter for next iteration $\theta_{t+1} := \theta_t - \eta \frac{1}{B} \sum_{i=1}^{B} (\nabla_\theta l(\theta_t, \mathbf{z}_t^{non}) + \nabla_\theta l(\theta_t, \mathbf{z}_t^{neg}))$;
26 **end**
27 output $\theta_T$ and compute overall privacy cost through moments accountant.

---

**Privacy Analysis.** In order to compose the transformed gradient perturbation and the direct isotropic gradient perturbation in each iteration for DP analysis, we introduce Mixed Multivariate Gaussian Analysis below.

**Theorem 2.** *Mixed Multivariate Gaussian Analysis. Let $\mathcal{G} : \mathbb{R}^v \to \mathbb{R}^w$ be an arbitrary $w$-dimensional function, and $\Delta_{\mathcal{G}} = \max_{\mathcal{D}, \mathcal{D}'} \|\mathcal{G}(\mathcal{D}) - \mathcal{G}(\mathcal{D}')\|_2$. Mixed Multivariate Gaussian Analysis is the mix of a Multivariate Gaussian Mechanism $\mathcal{M}_1$ with the covariance $\boldsymbol{\Sigma}_{(i)} \in \mathbb{R}^{w \times w}$ and a Gaussian Mechanism $\mathcal{M}_2$ with $\bar{\sigma}$ adding noise to each of the $w$ elements of the output. This mixed mechanism is $(\epsilon, \delta)$-DP, with*

$$\epsilon \in (0, 1], \sqrt{\bar{\sigma}^2 + TS_{min}(\mathbf{M}_{(i)})\sigma^2} \geq \xi_{up} \geq \sqrt{2\ln(1.25/\delta)}\Delta_{\mathcal{G}}/\epsilon.$$

*where $\bar{\sigma}$ is the isotropic gradient perturbation, $T$ is the number of training steps, $S_{min}(\mathbf{M}_{(i)})$ is the minimum singular value of $\mathbf{M}_{(i)}$ and $\boldsymbol{\Sigma}_{(i)} \triangleq \mathbf{M}_{(i)}\sigma^2$.*

Mixed Multivariate Gaussian Analysis (**MMGA**) can be leveraged to analyze privacy guarantee provided by transformed gradient perturbation and additional gradient perturbation. In the following of this section, we will prove Theorem 2 and show that **MMGA** can work with moments accountant to provide a tighter DP estimation and preserve $(\epsilon, \delta)$-DP for deep learning models with pre-trained classifiers.

**Privacy Analysis for Vanilla SGD.** In vanilla SGD, the algorithm randomly picks one sample at each iteration and feeds it into the model for optimization. Given the initial parameters $\theta_0$, iteration $t$, the parameters are updated as: $\theta_{t+1} = \theta_t - \eta \, \nabla_{\theta_t} l(\theta_t, \mathbf{z}_t^{non})$, where $\eta$ denotes the learning rate and $\mathbf{z}_t^{non}$ denotes one perturbed sample randomly picked at iteration $t$. In optimization process, the transformed perturbation $\mathbf{p}_t$ is slightly different from the that in Equation (3).

**Lemma 2.** *Given perturbed example $\mathbf{z}_t^{non} = \mathbf{x}_t^{non} + \mathbf{b}_t$ with $\mathbf{b}_t^{(k)} \sim \mathcal{N}(0, \sigma^2)$, the number of training steps $T$, and $C$-Lipschitz continuous loss $l$. The gradient $\nabla_{\theta_t} l(\mathbf{z}_t^{non}, \theta_t)$ at each step of vanilla SGD can be reformulated as:*

$$\nabla_{\theta_t} l(\mathbf{z}_t^{non}, \theta_t) = \nabla_{\theta_t} l(\mathbf{x}_t^{non}, \theta_t) + \mathbf{p}_t, \tag{4}$$

*where the transformed perturbation $\mathbf{p}_t \sim \mathcal{N}(0, TS_{min}(\mathbf{M}_t)\sigma^2\mathbf{I})$, $\mathbf{M}_t = \mathbf{A}_{(i)}\mathbf{A}_{(i)}^\mathsf{T}$, $\mathbf{A}_{(i)} = \mathbf{J}_{\theta_t} \nabla_{\mathbf{x}_{(i)}^{non}} l(\mathbf{x}_{(i)}^{non}, \theta_t)$.*

The detailed proof of Lemma 2 can be found in Appendix G. Theorem 2 can be proved by Lemma 2, the definition of $\bar{\sigma}$ and Theorem 1. Thus, we have the following corollary for "non-negative cases":

**Corollary 2.** *Given an input perturbation $\mathbf{b}_t \in \mathbb{R}^v$ with $\mathbf{b}_t^{(k)} \sim \mathcal{N}(0, \sigma^2)$, the transformation matrix $\mathbf{A}_t$ and additional gradient perturbation with scale $\bar{\sigma}$, the DP guarantee of the **MMGA** is equivalent to that of a **GM** with its perturbation following a Gaussian distribution $\mathcal{N}(0, \xi_{up}^2)$.*

The above analysis for "non-negative cases" leverages perturbation transformation and **MMGA** to analyze DP. For "negative cases", because the perturbation transformation is no longer required and the gradient perturbation with $\bar{\sigma} = \xi_{up}$ is directly added to $\nabla_\theta l(\theta_t, \mathbf{z}_t^{neg})$, the **MMGA** is equivalent to traditional **GM** with perturbation following $\mathcal{N}(0, \xi_{up}^2)$. Therefore, we can conclude that Corollary 2 is applicable to both "non-negative cases" and "negative cases".

**Privacy Analysis for Mini-batch SGD.** The above claims for Vanilla SGD can be adapted to Mini-batch SGD by setting $\mathbf{M}_t = \frac{1}{B^2} \sum_{i=1}^{B} \mathbf{A}_{(i)}\mathbf{A}_{(i)}^\mathsf{T}$ instead of $\mathbf{M}_t = \mathbf{A}_{(i)}\mathbf{A}_{(i)}^\mathsf{T}$ in Vanilla SGD. The detailed proof can be found in Appendix H.

**Tighter Composition via Moments accountant.** While Corollary 2 is derived with simple compositions, moments accountant can be applied with **MMGA** to provide a tighter DP composition for Algorithm 1.

**Theorem 3.** *There exist constants $c_1$ and $c_2$ so that given sampling probability $q = \frac{B}{N}$ and the number of training steps $T$, for any $\epsilon < c_1 q^2$, Algorithm 1 is $(\epsilon, \delta)$-differential private for any $\delta > 0$ if*

$$\xi_{up} \geq c_2 \frac{q\sqrt{T log(1/\delta)}}{\epsilon} \tag{5}$$

The proof of Theorem 8 can be found in Appendix I.

**Discussion.** We note that although the transformation matrix $\mathbf{A}_{(i)}$ requires the clean example $\mathbf{x}_{(i)}$, the calculation of $\mathbf{A}_{(i)}$ does not incur privacy cost for the denoiser. This is because the transformation process and the calculation of $\mathbf{A}_{(i)}$ is only for analyzing the DP of the input perturbation, i.e., the clean example $\mathbf{x}_{(i)}$ does not actually contribute to the gradient $\nabla_\theta l(\mathbf{z}_{(i)}^{non}, \theta)$. Another potential privacy concern is about the observations that "non-negative cases" and "negative cases" use different scales of additional gradient perturbation. Even for "non-negative cases", different data will be applied with different $\bar{\sigma}$. We claim that the different additional perturbation scales among "non-negative cases" and "negative cases" will not incur privacy violation, because we have proven that input perturbation also provides certain level of privacy guarantee, which is analyzed in a way that we transform it into gradient perturbation. We first use perturbation transform matrix and **MGM** to calculate how much gradient perturbation scale can be transformed from input perturbation for

different data. Then we add additional gradient perturbation onto transformed gradient perturbation to ensure that the overall gradient perturbation scale $\bar{\sigma}^2 + TS_{min}(\mathbf{M}_t)\sigma^2 \geq \xi_{up}^2$ for each data. On the one hand, the calculation of $\bar{\sigma}$ is not visible to users or attackers. On the other hand, different $\bar{\sigma}$ can ensure the overall scale $\bar{\sigma}^2 + TS_{min}(\mathbf{M}_t)\sigma^2$ is a consistent lower bound for all training data.

## 3 EXPERIMENTS

In this section, we will show our experiments on two benchmark datasets, MNIST and CIFAR-10. These experiments are conducted to prove that 1) TransDenoiser can provide high level of certified robustness through randomized smoothing, 2) the input perturbation transformation can save a considerable amount of DP budget and thus improve the model performance.

### 3.1 CONFIGURATIONS

**Baseline and ablation studies.** We employ SecureSGD (Phan et al., 2019) and StoBatch (Phan et al., 2020) two architectures in baseline approaches, and compare the certified robustness and DP performance on MNIST and CIFAR-10 datasets. We acquire two versions of SecureSGD through different training strategies: SecureSGD_sct is acquired by training an entire classifier from scratch, and SecureSGD_prt is acquired by training the denoiser and fixing the pre-trained classifier. For StoBatch, we only acquire the one training from scratch, because the denoiser with pre-trainied classifier can not fit to its architecture. We also conduct two ablation studies with two variants of TransDenoiser: 1) TransDenoiser_nodp which only contains input perturbation for certified robustness, without the perturbation transformation and additional gradient perturbation for DP; 2) TransDenoiser_sepdp which contains input perturbation for certified robustness and separate gradient perturbation for DP, without utilizing input perturbation via perturbation transformation.

**Models.** Pre-trained classifiers are trained on public datasets, and we use convolutional and transposed convolutional layers to build the autoencoder based denoiser for both MNIST and CIFAR10 datasets. The details of pre-trained classifiers and denoiser can be found in Appendix M.

**Adversarial examples.** We use four different attack algorithms, i.e., FGSM, I-FGSM (Kurakin et al., 2016), Momentum Iterative Method (MIM) (Dong et al., 2017), and MadryEtAl (Madry et al., 2017), to craft adversarial examples. These algorithms apply $l_2$-norm attack on the pre-trained classifier under a white-box setting. Given a threshold $L_{atk}$ of perturbation norm, adversarial example $\mathbf{x}'$ can be represented as $\mathbf{x}' = \mathbf{x} + \psi$ s.t. $\forall \psi \in \mathcal{R}^v, \|\psi\|_2 \leq L_{atk}$.

**Certification.** We employ the randomized smoothing of Cohen et al. (Cohen et al., 2019) in our work. A function $robustRadius(\mathbf{x}_{(i)}, \sigma)$ is designed to return a certified radius $\kappa$ given the input and its perturbation scale. This indicates that the randomized smoothed model is certified robust around $\mathbf{x}_{(i)}$ within the radius $\kappa$.

**Evaluation metrics.** We evaluate the performance in terms of certified accuracy (CertAcc) on clean examples and conventional accuracy (ConvAcc) on both clean and adversarial examples. $CertAcc = \frac{1}{N}isCorrect(\mathbf{x}_{(i)})(robustRadius(\mathbf{x}_{(i)}, \sigma) \geq R)$, $ConvAcc = \frac{isCorrect(\mathbf{x}'_{(i)})}{N}$ for adversarial example; $\frac{isCorrect(\mathbf{x}_{(i)})}{N}$ for clean example, where $N$ denotes the test data size, $\mathbf{x}'_{(i)}$ denotes the adversarial example, $isCorrect(\mathbf{x}_{(i)})$ denotes the function returning 1 when the prediction on $\mathbf{x}_{(i)}$ is correct and 0 otherwise, $isCorrect(\mathbf{x}'_{(i)})$ works the same on $\mathbf{x}'_{(i)}$, $robustRadius(\mathbf{x}_{(i)}, \sigma) \geq R$ returns 1 when the certified radius $\kappa$ is equal or larger than the threshold $R$ and 0 otherwise.

**Implementation details.** The detailed implementation and code can be found in Appendix M.

### 3.2 EXPERIMENTAL RESULTS

We conduct our experiments on MNIST and CIFAR-10 to show that TransDenoiser can simultaneously achieve both differential privacy and certified robustness via input and gradient perturbation.

For the following experiments, we will compare with 1) SecureSGD_sct, SecureSGD_prt and StoBatch to show that TransDenoiser achieves better performance than baselines on certified robustness

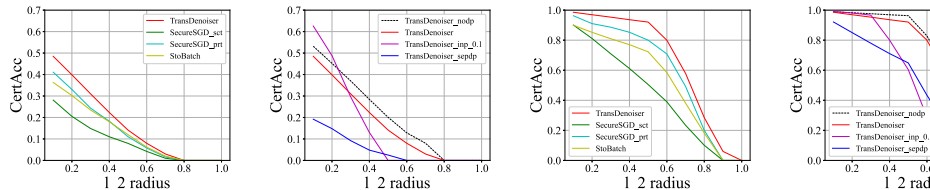

(a) TransDenoiser and baselines on CIFAR10 (b) TransDenoiser and ablation cases on CIFAR10 (c) TransDenoiser and baselines on MNIST (d) TransDenoiser and ablation cases on MNIST

Figure 2: Comparison among TransDenoiser, baselines and ablation cases for certified accuracy vs. $l_2$ radii on two datasets. The input perturbation scale = 0.25, the overall gradient perturbation scale = 2.0 ($\geq 2.0$ for TransDenoiser), and guarantee $(1.0, 1e-5)$-DP for private models.

and DP, 2) compare with TransDenoiser_nodp to show that TransDenoiser still achieves high level of certified robustness after adding gradient perturbation for DP, and 3) compare with TransDenoiser_sepdp to show that input perturbation transformation effectively saves a considerable amount of DP budget and improves the utility performance.

**Certified robustness.** We demonstrate that TransDenoiser can achieve high level of certified robustness on both MNIST and CIFAR10. We conduct experiments to measure the CertAcc on clean examples with different $l_2$ radii of different methods on the two datasets, shown in Figure 2a and Figure 2c. TransDenoiser significantly outperforms the state-of-the-art SecureSGD_prt, SecureSGD_sct and StoBatch algorithms on both datasets, thanks to the benefit of perturbation transformation and randomized smoothing.

Figure 2b and Figure 2d show the comparison with ablation cases. Besides TransDenoiser_nodp and TransDenoiser_sepdp that we already introduced in Sec 3.1, we add one more ablation case TransDenoiser_inp_0.1 denoting that the input perturbation scale for this TransDenoiser is 0.1 rather than 0.25. Comparing these ablation cases, TransDenoiser achieves similar CertAcc as TransDenoiser_nodp, which means that TransDenoiser needs to add very little additional perturbation for ensuring DP thanks to the benefit of input perturbation that is being exploited. Compared with TransDenoiser_sepdp that uses separate gradient perturbation for DP, TransDenoiser saves significant DP budget and thus requires less perturbation for DP, leading to significantly higher accuracy. Comparing with TransDenoiser_inp_0.1, we find that TransDenoiser_inp_0.1 can achieve highest CertAcc when $l_2$ radius is small, but it drops quickly as radius increases. This is because 1) smaller input perturbation scale will bring less randomization to the model, and thus improves performance; 2) smaller input perturbation scale can only defend against less "powerful" adversarial attacks, and thus CertAcc drops when attack radius increases. Comparing the two datasets, they show similar trend besides the fact that MNIST has higher accuracy for all methods in general due to its simplicity.

**Empirical defense.** Certified robustness shows the theoretical defense against adversarial examples, we also conduct experiments to show that TransDenoiser can empirically defend against adversarial examples from different attacks. We only show the results against FGSM, I-FGSM attacks here, the more detailed experiments can be found in Appendix M. Figure 7a and Figure 7c show the convAcc of TransDenoiser and baselines with respect to varying attack norm bound for different attack methods on two datasets, respectively. Compared with these baselines, TransDenoiser achieves better empirical performance with any attack norm bound of all attacks. Figure 7b and Figure 7d show the comparison of TransDenoiser and ablation cases. Compared with TransDenoiser_nodp, TransDenoiser achieves similar ConvAcc, which proves that perturbation for DP in TransDenoiser does not affect the ConvAcc too much. Compared with TransDenoiser_sepdp, TransDenoiser effectively saves DP budget and achieves better empirical performance against adversarial examples. In all these figures, we add a curve named "Clean examples" to represent ConvAcc that clean examples pass through TransDenoiser. As can be seen, ConvAcc for clean examples keep consistent as attack norm bound increasing, and TransDenoisercan achieve similar ConAcc as "Clean examples" when attack norm bound is small.

Observing the similar results between CertAcc and ConvAcc, we see that the certified accuracy on clean examples provides a good estimation for the empirical robustness of the model. If a model achieves relatively high CertAcc on clean examples, it can have a high probability to achieve high ConvAcc on adversarial examples.

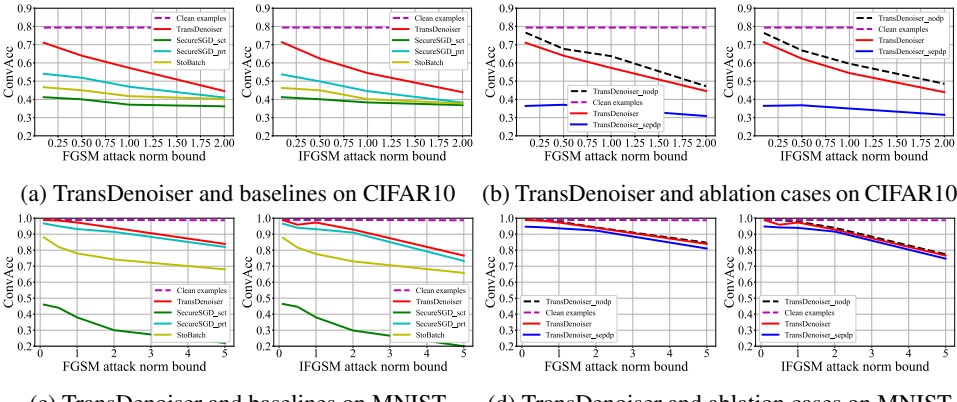

(a) TransDenoiser and baselines on CIFAR10    (b) TransDenoiser and ablation cases on CIFAR10

(c) TransDenoiser and baselines on MNIST    (d) TransDenoiser and ablation cases on MNIST

Figure 3: More comparison among TransDenoiser, baselines and ablation cases for conventional accuracy vs. $l_2$ radii on two datasets. The input perturbation scale on CIFAR10 = 0.1, on MNIST = 0.25, the overall gradient perturbation scale = 2.0 ($\geq$ 2.0 for TransDenoiser), and guarantee $(1.0, 1e-5)$-DP for private models.

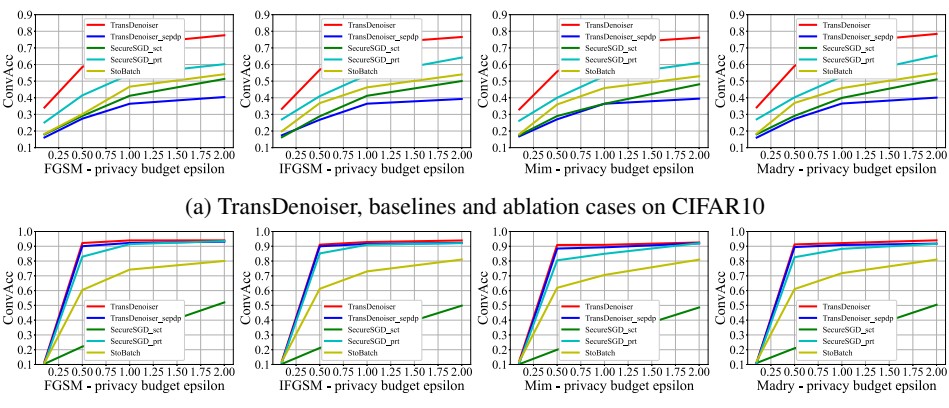

(a) TransDenoiser, baselines and ablation cases on CIFAR10

(b) TransDenoiser, baselines and ablation cases on MNIST

Figure 4: Comparison among TransDenoiser, baselines and ablation cases for conventional Accuracy vs. $\epsilon$ on two datasets. The input perturbation scale on CIFAR10 = 0.1, on MNIST = 0.25, attack norm bound = 2.0, the overall gradient perturbation scale = 2.0 ($\geq$ 2.0 for TransDenoiser), and $\delta = 1e-5$ for DP.

**Differential privacy.** We also evaluate the tradeoff between accuracy and privacy for different methods. As can be seen from Figure 4a and Figure 4b, given the same $\epsilon$, TransDenoiser can always achieve higher ConvAcc on different attacks similar to what we have observed so far. In addition, with increasing epsilon, all methods achive a higher accuracy as expected. On MNIST dataset, TransDenoiser_sepdp achieves similar result with TransDenoiser.

## 4  CONCLUSIONS AND FUTURE WORK

In this paper, we have proposed TransDenoiser to achieve both DP and certified robustness via input perturbation. TransDenoiser stands as the first attempt to achieve both for the vastly existing, yet under-studied, pre-trained model setting. We leverage input perturbation transformation to efficiently transform input perturbation into gradient perturbation. We propose **MGM** and **MMGA** to analyze DP of the transformed gradient perturbation and combine **MMGA** with moments accountant to provide a tight bound on DP guarantee. Therefore, TransDenoiser effectively saves a considerable DP budget and improves the utility performance compared to using gradient perturbation independently to achieve DP. Our experiments on two benchmark datasets verify the performance advantage of TransDenoiser w.r.t. both DP and certified robustness compared to state-of-the-art methods. In future work, we plan to utilize more advanced DP analysis approach, e.g., analytical moments accountant (Balle & Wang, 2018), to derive a tighter bound on DP and further improve the privacy and utility tradeoff.

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
