# OpenReview forum: "Two Birds, One Stone:  Achieving both Differential Privacy and Certified Robustness for Pre-trained Classifiers via Input Perturbation"
_ICLR.cc/2022/Conference — ICLR 2022 Submitted_

### Official Review · Reviewer_s8jm · 2021-11-01

**Correctness:** 2
**Technical Novelty And Significance:** 2
**Empirical Novelty And Significance:** 2
**Recommendation:** 3
**Confidence:** 4

**Main Review:**

**Strengths**

Privacy and robustness to adversarial examples are two hot topics within the ML community, especially when considering large models for image or speech recognition. Therefore, I believe that the main focus of this paper, i.e., the integration of these two notions for pre-trained classifiers, is very relevant to the ICLR community. Also, the main point of the article is quite simple and easy to understand from a high-level perspective. Finally, the idea of trying to translate the input noise injection into gradient perturbation to simplify the privacy analysis seems interesting.

**Weaknesses**

My main concern is with the technical quality of the article. In fact, I am not sure that the claims of the paper are technically correct, especially with respect to Lemmas 1 and 2. Although, from a general perspective, the concepts of input noise and gradient perturbation seem related, in my opinion, neither Lemma 1 nor Lemma 2 demonstrate a clear connection.  I provide some details below.

1. Lemma 1 states that for a certain type of perturbed examples $z_{(i)}^{non}$ (defined in section 2.2), the gradient computed at $z_{(i)}^{non}$ can be lower bounded by the gradient computed at the initial point $x_{(i)}^{non}$ plus some noise that depends on the Jacobian matrix of the loss function at $x_{(i)}^{non}$. First, the statement itself appears to be very confusing to me because the authors compare two random vectors (with infinite support) without explaining the meaning of the term $\geq$. Second, the analysis that the authors provide by stating that, according to Lemma 1, "the DP guarantee provided by the perturbation of the transformed gradient is the lower bound of the one provided by the perturbation of the input" lacks justification, especially since the lemma only holds for a specific family of perturbed inputs.  Finally, looking at the proof, I have some additional concerns, among which a) the reason for the jump from (9) to (10) is not clear to me, and b) the transition from (10) to the equality between the gradient of $z_{(i)}$ and the perturbed gradient of $x_{(i)}$ is also not clear to me.
2. Lemma 2 provides a similar statement with an equality but was not provided with a formal proof. Instead, the authors present another lemma in the appendices (Lemma 3, also without a proof), which is very similar to Lemma 1 and claim that Lemma 2 can be derived from Lemma 3. This claim does not seem sufficiently conclusive to me.

Finally, I have the impression that the paper claims too much its technical contribution on the analysis of multivariate Gaussian noise injection. In fact, as I understand it, this work can be considered as a special case of previous work studying matrix-valued Gaussian mechanisms [2]. I think the authors should compare with this previous work.

**Additional comments and questions**

In experiments, I am not sure that the comparison of TransDenoiser with previous methods is fair in terms of privacy preservation. As I understand it in [3] and [4], the model is directly learned with differential privacy, thus protecting the dataset used to learn the model. However, in this paper, the authors only claim to preserve privacy on the fine-tuning dataset, thus leaving the dataset used in the pre-trained model unprotected. I have two concerns with this: a) the authors are comparing methods that do not preserve privacy on the same dataset, which makes the comparison unfair compared to previous methods, and b) I checked the pre-trained classifiers and it appears that they use the same datasets as the authors' trained auto-encoder (since these models are not trained with privacy, I think this represents a clear privacy breach).

In Thereom 3, the authors present a result on privacy preservation for Algorithm 1 based on a previous result in [5]. This result is only valid if we consider algorithms that use Poisson sampling to select the mini-batch at each round. However, Algorithm 1 does not seem to be using Poisson sampling since the size of the mini-batch is constant and equal to B (see line 25 of the algorithm).

I think the presentation of the technical contribution could be improved. The appendix presents several statements re-demonstrating existing results in the literature on privacy and adversarial robustness (Theorem 5, 6 and Appendix B). From my point of view, these do not help the overall understanding of the contributions of the paper. I suggest presenting only the proofs of the original contributions in the appendix, and simply citing the existing papers for the earlier work when needed.


[2] MVG Mechanism: Differential Privacy under Matrix-Valued Query Thee Chanyaswad, Alex Dytso, H. Vincent Poor, Prateek Mittal

[3] Heterogeneous Gaussian Mechanism: Preserving Differential Privacy in Deep Learning with Provable Robustness NhatHai Phan, Minh Vu, Yang Liu, Ruoming Jin, Dejing Dou, Xintao Wu, My T. Thai

[4] Scalable Differential Privacy with Certified Robustness in Adversarial Learning NhatHai Phan, My T. Thai, Han Hu, Ruoming Jin, Tong Sun, Dejing Dou

[5] Deep Learning with Differential Privacy Martín Abadi, Andy Chu, Ian Goodfellow, H. Brendan McMahan, Ilya Mironov, Kunal Talwar, Li Zhang


**Summary Of The Paper:**

This paper studies the problem of integrating differential privacy and robustness to adversarial examples for pre-trained machine learning models. Specifically, this work aims at designing methods that guarantee both privacy and robustness without having to re-train the model at hand. To achieve this goal, the authors build upon an existing technique in the adversarial example literature that involves placing a denoising auto-encoder in front of a pre-trained model before applying a noise injection scheme known as "randomized smoothing" [1]. While this technique is known to provide state-of-the-art "certified accuracy" against adversarial examples, its privacy guarantees remained to be studied. This work proposes to do just that by adapting the algorithm to guarantee differential privacy for the dataset used to train the auto-encoder. The authors claim three main contributions:

1. Exploiting the intrinsic train-time input perturbation that existed in the previous implementation of the algorithm and composing it with an explicit gradient perturbation to satisfy differential privacy. The authors claim that their treatment of this input perturbation allows a finer analysis of the algorithm's privacy, which ultimately leads to better accuracy, for the same privacy guarantees.
2. Introducing two new analytical tools, namely MGM and MMGA,  for analyzing the privacy guarantees of multivariate Gaussian noise injection.
3. Conducting extensive experiments on several benchmark datasets to demonstrate that their algorithm, called « TransDenoiser »,  provides better privacy guarantees and achieves similar level of certified robustness compared to previous works.

[1] Provably Robust Deep Learning via Adversarially Trained Smoothed Classifiers Hadi Salman, Greg Yang, Jerry Li, Pengchuan Zhang, Huan Zhang, Ilya Razenshteyn, Sebastien Bubeck


**Summary Of The Review:**

While I think this article studies an interesting problem, I do not think it presents its contributions convincingly enough. In particular, I have concerns about the technical quality and the novelty of the article that lead me to recommend its rejection.

---

> ### Author Response · Authors · 2021-11-23
> **Responses to comments and suggestions**
>
> Thank you very much for your review and comments. We have carefully read through these comments and have explanations as follows.
>
> For weaknesses:
>
> 1.1 Although x_(i)^non and z_(i)^non are two random vectors, the input perturbation, z_(i)^non – x_(i)^non, is deterministic for each batch given input perturbation scale sigma. We calculate the higher-order terms by using this deterministic perturbation (line 7,8 in algorithm 1), and thus can derive the inequality.
>
> 1.2 The proof of Lemma 1 in Appendix E proves that the right-hand side is the lower bound of the left-hand side. We recognize that we missed the clarification of the specific family of perturbation in the paper. We will clarify that all the perturbations introduced in our work are Gaussian perturbations.
>
> 1.3 a) The reason that we go from (9) to (10) is that we need to build a connection between input perturbation and gradient perturbation. The former is easy to obtain by subtraction between z and x, while the latter has to be obtained by calculating the gradient of the loss.
> b) We will add more steps in (10) to show how we get the final equation.
>
> 2. We will re-write the proof of lemma 2 with a clearer explanation.
>
> 3.  We did miss the comparison between our work and MVG. We will have a detailed discussion about it in the revisions: please refer to the response to Reviewer K9zu(https://openreview.net/forum?id=keQjAwuC7j-&noteId=KWqlkdEZbDc) and the first point to Reviewer t4Xc(https://openreview.net/forum?id=keQjAwuC7j-&noteId=iqZyJiNqCLx)
>
> For additional comments and questions:
>
> 1. It is a fair comparison. Works in [3] and [4] are not in the pre-trained model settings. However, in our experiments, we adapt them to the pre-trained model setting and compare results. We will have a clearer explanation about this adaptation.
>
> 2. a) It is a fair comparison. All models run on the same training dataset and validate on the same validation dataset.
>
> b) In our experiments, we randomly split the dataset into two parts: public and private. The public dataset is used to train the pre-trained classifier, and the private one is used to train the denoiser. We only protect the DP of the private dataset.
>
> 3. According to Algorithm 1 in Section 3.1. of [5], the algorithm utilizes random sampling from the training dataset to get a batch of samples at each iteration. That is exactly what we do.
>
> 4.  Thanks for your suggestions. We will follow these in the revisions.
>
> [2] MVG Mechanism: Differential Privacy under Matrix-Valued Query Thee Chanyaswad, Alex Dytso, H. Vincent Poor, Prateek Mittal
>
> [3] Heterogeneous Gaussian Mechanism: Preserving Differential Privacy in Deep Learning with Provable Robustness NhatHai Phan, Minh Vu, Yang Liu, Ruoming Jin, Dejing Dou, Xintao Wu, My T. Thai
>
> [4] Scalable Differential Privacy with Certified Robustness in Adversarial Learning NhatHai Phan, My T. Thai, Han Hu, Ruoming Jin, Tong Sun, Dejing Dou
>
> [5] Deep Learning with Differential Privacy Martín Abadi, Andy Chu, Ian Goodfellow, H. Brendan McMahan, Ilya Mironov, Kunal Talwar, Li Zhang

---

### Official Review · Reviewer_wLgg · 2021-11-02

**Correctness:** 3
**Technical Novelty And Significance:** 2
**Empirical Novelty And Significance:** 2
**Recommendation:** 3
**Confidence:** 4

**Main Review:**

Strengths:
* The idea is well described.
* The authors provided detailed analysis both theoretically and experimentally.

Weaknesses / discussion questions:
* The proposed TransDenoiser is lack of novelty. This paper mainly builds off the work by (Salman et al, 2020) [1], which proposed to train a denoiser on input perturbations and leveraged randomized smoothing to achieve certified robustness. The main difference is that in TransDenoiser, additional gradient perturbations are generated. However, the comparison with [1] is missing. The lack of comparison with the most relevant baseline reduces the confidence.

* Throughout the paper, it is not made clear why achieving DP is important, and what is the difference between "partial" and "overall" DP. I think it would be good to include some brief background on DP in the related work section instead of in the appendix. A good chunk of your introduction could be moved to the related work section as well.

* In terms of clarity, the overall writing could be greatly improved. There are several typos, confusing sentences and symbol choices (I listed several in minor issues). Specifically, it’s hard to follow your TransDenoiser Training Algorithm.

* Does the proposed method work for L-infinity norm as well, both theoretically and experimentally?

* The experimental results are not entirely convincing to me. For a thorough evaluation, it would be better to report robust accuracy against PGD (using MadryEtAl is very confucing...), CW and Auto Attack. My specific concerns are the following:
  * All the methods should be given with a better name, current versions, like "xxx_sct", "xxx_prt", "xxx_sepdp" are not easy to follow.
  * The captions of Figure 2 are mixed together and confusing.
  * I’m not sure what's going on in Figure 3
    * there is no figure reference in main text. I just suppose the corresponding explanations are under "Empirical defense", please correct me if I am wrong.
    * only 'clean example' for TransDenoiser is provided, what about other methods?
    * besides, the better robustness may be caused by the trade-off between natural accuracy and robustness, with lack of the 'clean example' given by other baselines, to me, it is not convincing to directly draw the conclusion that "the certified accuracy on clean examples provides a good estimation for the empirical robustness of the model".

Minor:
* I found authors make their statements misleading. For example,
  * in page 2, the paper says "compared with [1], TransDenoiser can .... without retraining the pre-trained models", however [1] fixed the pre-trained models instead of retraining.
  * in page 3, the paper says "different from [1], ..., the objective function we use to optimize the denoiser contains the standard reconstruction MSE", however [1] also used MSE.
* Under "Empirical defense", the authors are mainly describe figures in the Appendix.
* The proposed methods should be evaluated on larger datasets (e.g., CIFAR-100) and more "popular" models (e.g., ResNet-xx, WRN-xx) to demonstrate its effectiveness thoroughly.


**Summary Of The Paper:**

In this paper, authors studied the problem of achieving both the overall differential privacy and certified robustness simultaneously for pre-trained models. They proposed a framework called TransDenoiser based on an existing framework (Salman et al, 2020) [1] by adding additional and transformed gradient perturbations for the overall DP. Authors analyzed DP guarantee provided by these perturbations and empirically evaluate their methods on MNIST and CIFAR-10, and shown that TransDenoiser is effective against FGSM and PGD attacks with guaranteed DP.

**Summary Of The Review:**

This paper mainly builds off the work by (Salman et al, 2020) [1]. Although DP analysis and tighter bound on DP guarantee are of some significance, the authors are suggested to 1) compare their proposed method with [1], 2) improve overall writing clarity, and 3) significant improvements over experiment settings.

---

> ### Author Response · Authors · 2021-11-23
> **Responses to comments and suggestions**
>
> Thank you very much for your review and comments. We have carefully read through these comments and have explanations as follows.
>
> For weakness:
>
> 1. We actually plotted the curve for only achieving certified robustness (black dotted line in Figure 2b, 2d), which is exactly the results from (Salman et al, 2020). As can be seen from the figures, the performance of the model only achieving certified robustness is slightly better than TransDenoiser. The slight degradation of TransDenoiser comes from the fact that it adds additional gradient perturbation to guarantee DP. In the revision, we will make the notation and explanation more clear.
>
> 2. Thanks for your suggestions! We will move some of the preliminary and related work from Appendix A to the main paper in the revision. We claim that achieving DP is important, because privacy leakage of training data and adversarial attacks are two main vulnerabilities of deep learning models. Most existing works focus on one of these two challenges. A few works consider both, but they address these two challenges independently without exploring the connection between using randomization to achieve DP and certified robustness.
>
> 3. We will update the paper and have a clearer explanation about the training algorithm.
>
> 4. We leverage a state-of-the-art certified robustness methodology which only provides L2-norm robustness. The Linfinity norm would be an interesting future research direction.
>
> 5. We will add experiments for C&W and Auto attack in the revision. We will rename these baseline and ablation models with clearer explanations.
>
> Figure 3 is for empirical defense, we will fix the bug here.
>
> We will add figures to show the conventional accuracy of clean examples on different models. In our experimental results, clean examples achieve the highest conventional accuracy after passing through TransDenoiser compared to other models.
>
> For minor:
>
> We will have clearer explanations in the revisions.
>
> 1. Statement on page 2: What we want to emphasize is that we can achieve both DP and certified robustness while [1] only achieves certified robustness. And yes, both our work and [1] fix the pre-trained models.
>
> 2. Statement on Page 3: We miss “only” her, i.e., we only use MSE without stability objective in [1].
>
> 3. There are errors in the figure references. We will correct them in the revisions.
>
> 4. We will conduct more experiments, e.g., CIFAR-100 with resnet-50, in the revisions

---

### Official Review · Reviewer_K9zu · 2021-11-03

**Correctness:** 4
**Technical Novelty And Significance:** 3
**Empirical Novelty And Significance:** 3
**Recommendation:** 6
**Confidence:** 4

**Main Review:**

strengths:
1). This paper considers transforming input perturbation into gradient perturbation, then the noise introduced by random smoothing can be quantified with the explicit gradient perturbation for the privacy guarantee.
2). To analyze the privacy guarantee, Multivariate Gaussian mechanism is proposed by considering multivariate Gaussian perturbation
3). The proposed method are conducted on various datasets and adversial attacks to show the effectiveness.

weaknesses:
1), Multivariate Gaussian Mechanism is not new, and many previous works are also investigated multivariate Gaussian differential privacy to achieve DP. For example, Chanyaswad et al in [1] proposed a MVG mechanism, which adds a matrix-valued noise drawn from a matrix-variate Gaussian distribution, and also introduce the directional noise in MVG that can further imporve the utility. Further, Yang et al in [2] proposed a Matrix Gaussian Mechanisms for matrix value with better utility.


[1] Chanyaswad, Thee, Alex Dytso, H. Vincent Poor, and Prateek Mittal. "Mvg mechanism: Differential privacy under matrix-valued query." In Proceedings of the 2018 ACM SIGSAC Conference on Computer and Communications Security, pp. 230-246. 2018.
[2] Yang, Jungang, Liyao Xiang, Jiahao Yu, Xinbing Wang, Bin Guo, Zhetao Li, and Baochun Li. "Matrix Gaussian Mechanisms for Differentially-Private Learning." IEEE Transactions on Mobile Computing (2021).


**Summary Of The Paper:**

This paper focuses on providing both differential privacy and certified adversarial robustness to machine learning models.
The authors propose an algorithm called TransDenoiser to achieve such a goal. TransDenoiser consists a denoiser through both input and
gradient perturbation for achieving DP and certified robustness, and following by a pre-trained classifier for classification. The privacy guarantee is carefully analyzed. Extensive experiments demonstrate the effectiveness of proposed method from model utility and adversarial robustness.

**Summary Of The Review:**

This paper investigates how the random smoothing noise can be transformed into gradient perturbation, and then carefully compute the privacy loss, which seems an interesting method.

---

> ### Author Response · Authors · 2021-11-23
> **Responses to comments and suggestions**
>
> Thank you very much for your review and comments. We have carefully read through these comments and have explanations as follows.
>
> For weakness:
>
> Our Multivariate Gaussian Mechanism can be regarded as a specific type of matrix-valued Gaussian Mechanism (MVG) in [1]. However, there are several differences between our work and [1]. [1] focuses on DP guarantee of matrix-valued query, while our work focuses on deep learning models. These two settings are quite different: in deep learning setting, the perturbation is added to each iteration during training, while in query setting, the perturbation is added only for a few times. Compared with MVG,  DP analysis of MGM in our work has less time cost, because it only requires the calculation of minimum singular value, while MVG requires all singular values and harmonic numbers. On the other hand, MGM is a theoretical mechanism in our work. We also propose MMGA to analyze the DP guarantee for empirical deep learning training algorithm, and MMGA is never introduced in other works.
>
> [2] proposes two specific types of MVG and proves that the utility performance of these mechanisms is better than MVG. However, these two mechanisms can not be applied to our framework. Because the transformed gradient perturbation in our work follows a multivariate Gaussian distribution that is neither of these two cases.
> In the revisions, we will add two references: MVG [1] and Matrix Gaussian Mechanisms [2] and have detailed discussions about them.
>
>
> [1] Chanyaswad, Thee, Alex Dytso, H. Vincent Poor, and Prateek Mittal. "Mvg mechanism: Differential privacy under matrix-valued query." In Proceedings of the 2018 ACM SIGSAC Conference on Computer and Communications Security, pp. 230-246. 2018.
>
> [2] Yang, Jungang, Liyao Xiang, Jiahao Yu, Xinbing Wang, Bin Guo, Zhetao Li, and Baochun Li. "Matrix Gaussian Mechanisms for Differentially-Private Learning." IEEE Transactions on Mobile Computing (2021).

---

### Official Review · Reviewer_t4Xc · 2021-11-03

**Correctness:** 4
**Technical Novelty And Significance:** 2
**Empirical Novelty And Significance:** 2
**Recommendation:** 5
**Confidence:** 3

**Main Review:**

Strengths:

It may be desirable to defend the same classifier against adversarial input perturbations and privacy attacks, so approaches which simultaneously guarantee certified robustness and differential privacy are interesting and useful.

Prior work has noted that certified robustness techniques provide differential privacy, current work provides a quantification of this claim.

Experiments show gain over prior methods.


Weaknesses:

The proposed perturbation mechanism is a simple combination of input and gradient noise, the transformation process is only for analyzing the Differential Privacy guarantee and involves a simple Taylor estimate and weakly uses the properties of the loss function.

The multivariate gaussian mechanism is a simple generalization of known mechanism in prior work.


Questions:

It is not clear what is the proportion of negative and non-negative examples, or what effect this proportion has on differential privacy.

How are the perturbation scale thresholds xi_low and xi_up set in the experiments?

Can gradient perturbation be useful for certified robustness?

Experiments compare with other approaches that simultaneously try certified robustness and DP, but what is the gap from approaches that optimize for exactly one or the other?


Other suggestions/comments:

Formal statements are sometimes unclear or without sufficient detail. E.g. what is the domain of input x_(i).

The readability of the paper would benefit from a brief consolidated notation/terminology section, e.g. defining (epsilon,delta)-dp

Theorem 1 simultaneously defines Multivariate Gaussian Mechanism and proves a property for it, would be clearer to separate out a complete formal definition.

In equation (2) is o the little-oh notation? It appears to not be the case in Algorithm 1? A clarification would be useful if there is abuse of notation, or if the little-oh choice is just coincidental.

In abstract and elsewhere the DP bounds with Moments accountant are said to be 'tight' but perhaps better to just say relatively tighter.

**Summary Of The Paper:**

The paper studies differential privacy for pre-trained certifiers that offer certified robustness through input perturbation. The key insight is to analyze differential privacy afforded by the input perturbation by noise transformation (computing corresponding gradient noise), and augmenting with gradient perturbation when needed for differential privacy guarantees. This improves upon past work by reducing the differential privacy budget needed by showing that some differential privacy is obtained from the input perturbation. In addition to new analysis techniques for providing the privacy bounds, experiments show gains over prior methods, e.g. better differential privacy under adversarial attack.

**Summary Of The Review:**

The paper considers an interesting and relevant question but the technical novelty and significance of the provided results does not seem to be sufficient. Algorithmic additions over prior work include using a combination of input and gradient noise and adding general multivariate Gaussian noise. The key analytic insight is to use Taylor approximation to estimate DP afforded by input noise. It is not clear how to set the hyperparameters of the proposed algorithm. Also formal presentation is somewhat lacking. Even though the approach beats some prior works empirically, I believe the work is marginally below acceptance threshold due to above issues.

---

> ### Author Response · Authors · 2021-11-23
> **Responses to comments and suggestions**
>
> Thanks for your review and comments. We have carefully read through these comments and have explanations as follows.
>
> For weakness:
> 1. We disagree that “the proposed perturbation mechanism is a simple combination of input and gradient noise. “
> While the high-level idea may be intuitive and straightforward, the main challenge is that the input and gradient noise cannot be simply added.  To address that, we 1) transform the input perturbation into gradient perturbation by Taylor expansion, 2) we propose Multivariate Gaussian Mechanism (MGM) to analyze the DP guarantee provided by the correlated transformed gradient perturbation, and 3) we propose Mixed multivariate Gaussian Analysis to analyze the DP guarantee provided by both transformed and additional gradient perturbation. Each of these steps is non-trivial and requires significant mathematical development and algorithmic proofs. We also conducted an extensive experimental study showing the feasibility and benefit of the approach.
>
> 2. We partially agree that “multivariate gaussian mechanism is a generalization of known mechanism in prior work”.
> Our Multivariate Gaussian Mechanism (MGM) can be regarded as a specific type of matrix-valued Gaussian (MVG) Mechanism in [1]. However, there are several differences between our work and [1]. [1] focuses on DP guarantee of matrix-valued query, while our work focuses on deep learning models. These two settings are quite different: in deep learning setting, the perturbation is added to each iteration during training, while in query setting, the perturbation is added only for a few times. Compared with MVG,  DP analysis of MGM in our work has less time cost, because it only requires the calculation of minimum singular value, while MVG requires all singular values and harmonic numbers.
> In addition,  MGM is a theoretical mechanism in our work. We also propose Mixed multivariate Gaussian Analysis (MMGA) to analyze the DP guarantee for empirical deep learning training algorithms, which has not been introduced in other works.
> In the revisions, we will add [1] and Matrix Gaussian Mechanisms [2] as references. [2] proposes two specific types of MVG and proves that the utility performance of these mechanisms is better than MVG. However, these two mechanisms can not be applied to our framework. Because the transformed gradient perturbation in our work follows a multivariate Gaussian distribution that is neither of these two cases.
>
>
> [1] Chanyaswad, Thee, Alex Dytso, H. Vincent Poor, and Prateek Mittal. "Mvg mechanism: Differential privacy under matrix-valued query." In Proceedings of the 2018 ACM SIGSAC Conference on Computer and Communications Security, pp. 230-246. 2018.
>
> [2] Yang, Jungang, Liyao Xiang, Jiahao Yu, Xinbing Wang, Bin Guo, Zhetao Li, and Baochun Li. "Matrix Gaussian Mechanisms for Differentially-Private Learning." IEEE Transactions on Mobile Computing (2021).
>
> For Questions:
> 1. The proportion of negative and non-negative cases represents how much gradient perturbation can be transformed from input perturbation. The higher the proportion of non-negative cases is, the more gradient perturbation can be transformed. More transformed gradient perturbation means less additional gradient perturbation is required and hence better utility performance. In our experiments, we find that different datasets and model architectures will affect the proportion and thus affect the utility performance. We will add these results in the revisions.
>
> 2. xi_low and xi_up are hyper-parameters defined before experiments, and xi_up determines the DP level we want to achieve, while xi_low determines the threshold we add additional gradient perturbation.
>
> 3. Intuitively, gradient perturbation should provide some robustness, since it introduces randomization to a model. However, it is non-trivial to derive the exact level of certified robustness, if it can be certified, given gradient perturbation.  This is an interesting and complementary problem to our work (DP--> certified robustness vs. certified robustness --> DP), which we will discuss in our future work.
>
> 4. We actually plotted the curve for only achieving certified robustness (black dotted line in Figure 2b, 2d).  We will use an explicit notation and explanation in the revision to make it more clear. We will also add the curve for only achieving DP.
>
> For suggestions/comments:
>
> We will carefully follow your suggestions to make the paper more readable and clearer, including the definition of the domain, the notation section, more explanation of MGM, clarification of the little-o notation and the updates of moments accountant claim.

---

### Decision · Program_Chairs · 2022-01-20

**Decision:**

Reject

**Comment:**

This paper develops a technique to provide both privacy and robustness
at the same time using differential privacy.

Unfortunately the paper in its current form does not have meaningfully
interpretable security or privacy claims. The reviewers point at a number
of these flaws that the authors do not address to the satisfaction of
the reviewers, but there are a few others as well.
- What is actually private, at the end of this whole procedure? If the
  actual "pretrained classifier" is not made private, then what's the
  purpose of the entire privacy setup in this paper? Why does the denoiser
  need to be private if the classifier isn't?
- The proof of Lemma 1 appears incorrect. The proof in Appendix E says that
  Equation 10 is true, but this sweeps all of the remaining Taylor series
  terms under the rug and doesn't deal with them. How are they handled?
- In Figure 4(a), what does it even mean to have a "FGSM privacy budget
  epsilon"? Or a "MIM privacy budget epsilon"? A privacy budget is almost
  always something defined with respect to the *training data privacy*,
  how does this relate to the attack in this paper?
- How does this paper compare prior *canonical* defenses, both on the
  robustness and privacy side? In particular, comparisons to adversarial
  training on the robustness side, and some recent DPSGD result on the
  privacy side?